# Generalizing End-To-End Autonomous Driving In Real-World Environments Using Zero-Shot LLMs

**Zeyu Dong[†], Yimin Zhu[†], Yansong Li[‡], Kevin Mahon[§], Yu Sun[§*]**
[†]Stony Brook University, [‡]University of Illinois Chicago, [§]Sunrise Technology Inc.

**Abstract:** Traditional autonomous driving methods adopt a modular design, decomposing tasks into sub-tasks, including perception, prediction, planning, and control. In contrast, end-to-end autonomous driving, directly outputs actions from raw sensor data avoiding error accumulation. However, training an end-to-end model requires a comprehensive dataset. Without adequate data, the end-to-end model exhibits poor generalization capabilities. Recently, large language models (LLMs) have been applied to enhance the generalization capabilities of end-to-end driving models. Most studies explore LLMs in an open-loop manner, where the output actions are compared to those of experts without direct activation in the real world. Other studies in closed-loop settings examine their results in simulated environments. In comparison, this paper proposes an efficient architecture that integrates multimodal LLMs into end-to-end real-world driving models in a closed-loop setting. The LLM periodically takes raw sensor data to generate high-level driving instructions. In our architecture, LLMs can effectively guide the end-to-end model, even at a slower rate than the raw sensor data, because updates aren't needed every time frame. This architecture relaxes the trade-off between the latency and inference quality of the LLM. It also allows us to choose from a wide variety of LLMs to improve high-level driving instructions and minimize fine-tuning costs. Consequently, our architecture reduces the data collection requirements because the LLMs do not directly output actions, and we only need to train a simple imitation learning model to output actions. In our experiments, the training data for the end-to-end model in a real-world environment consists of only simple obstacle configurations with one traffic cone, while the test environment is more complex and contains multiple obstacles placed in various positions. Experiments show that the proposed architecture enhances the generalization capabilities of the end-to-end model even without fine-tuning the LLM.

**Keywords:** End-to-end Autonomous Driving, Large Vision-Language Model, Generalization

## 1 Introduction

Traditional autonomous driving methods [1] adopt the module design pattern, that is, the task of autonomous driving is decomposed into several subtasks: perception [2, 3, 4], prediction [5, 6, 7], planning [8, 9, 10], and control [11, 12]. In contrast, end-to-end autonomous driving [13] takes raw sensor data directly and outputs action constraints for real-world inference.

The end-to-end model differs fundamentally from the module design in the context of training, i.e., it only requires sensor data labeled by expert actions instead of intermediate labels of the environment [14, 15, 16] (e.g., lanes, traffic lights, etc.) or other vehicles [17, 18, 19] (e.g., speed, relative locations, driving patterns, etc.). These intermediate labels are much harder to obtain in practice.

---

[*]Corresponding author: `yu.sun@sunriseaitech.com`

8th Conference on Robot Learning (CoRL 2024), Munich, Germany.

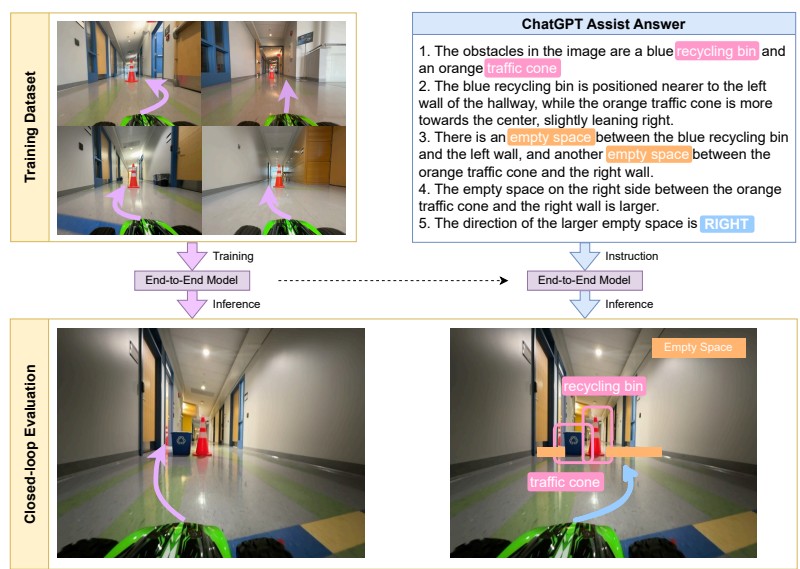

Figure 1: General idea of the architecture. *Top left*: the training dataset for the end-to-end model. The purple line represents the trajectories of the car for this route. *Top right*: an answer generated by `ChatGPT-4o` using the image in the bottom right. *Bottom left*: closed-loop evaluation on unseen scenarios for end-to-end model without LLMs. *Bottom right*: evaluation on unseen scenarios with high-level instructions from LLMs. The LLM still recognizes the new blue trash bin obstacle not included in the training dataset, evaluate viable empty space, and choose justifiable instructions.

In addition, end-to-end methods can potentially avoid error accumulation [13] caused by cascading modules in a modular design. End-to-end methods rely on data-driven approaches that require vast amounts of data to generalize effectively across diverse environment settings. Therefore, many previous end-to-end models have opted to gather extensive training data from a wide range of environments, often collected in simulators [20, 21]. Despite this, challenges persist when scaling from simple to complex scenarios that involve planning and reasoning, as discussed in Appendix A.

To address this issue, many recent works have explored the combination of language models with autonomous driving. It has been shown [22, 23] that small language models, for example, `Bert` or `GPT`, lack the generalization capability. In other words, they cannot generate future optimal trajectories based on the raw data for a new situation they have never encountered. Thus, Large Language Models (LLMs), especially multimodal LLMs, have been explored in autonomous driving [22, 23, 24] and have shown extraordinary abilities to understand driving environments and properly reason about future actions. Multimodal LLMs offer a notable generalization capability when pre-trained on a mixture of multimodal datasets. Compared to training in datasets designed for a specific task, multimodal datasets encompass a vast amount of diversity and have been shown to significantly enhance LLM performance across different tasks [25].

Only a handful of studies have applied LLMs to autonomous driving in a closed-loop manner. Due to the physical constraints and liability of real-world driving, these researchers have to design, test, and evaluate driving models in simulated environments that are not sensitive to the issue of the slow response time of LLMs. Furthermore, these studies require fine-tuning the LLMs using a significant amount of simulation data, which is impractical in real-world settings. Among these efforts, a direct approach to adopting LLMs in end-to-end autonomous driving is to employ them as the end-to-end model [26], that is, the LLM takes human instructions and outputs low-level actions.

An alternative approach uses LLMs to process the raw sensor data directly and generate actions. Many works explore this setting in an open-loop environment where LLM outputs trajectories [27] or steering/throttles signals [28]. These outputs are compared with an expert's trajectory instead of

being applied to the real world or simulated environment for feedback control. Other efforts rely on LLM in autonomous driving and adopt prompt engineering [29, 26, 30] instead of fine-tuning with LLM to guide the language model to generate the desired outputs.

More recently, Shao et al. [31] (LMDrive), Azarafza et al. [32], and Paul et al. [21] fine-tuned multimodal LLMs [33] and introduced the tuned LLMs to end-to-end autonomous driving in a closed-loop manner. In these approaches, the fine-tuned LLM takes raw sensor data and outputs future trajectories. One challenge of this approach is that the LLM model has a long inference latency, and if every action of the vehicle needs one LLM inference, it is impossible to use an LLM for real-world driving. However, slow LLM inference is not an issue for LMDrive because it uses the CARLA simulation environment. Another potential drawback of fine-tuning an LLM as the controller is *catastrophic forgetting* [34], a property that the LLM may fit the new training data and potentially lose its generalization properties due to fine-tuning.

In this paper, we tackle these issues by liberating the LLM model from direct action controls and creating an architecture that integrates multimodal LLMs with end-to-end autonomous driving. We will assess the effectiveness of the proposed method in a closed-loop real-world environment. In our architecture, the multimodal LLM takes the multimodal sensor data and outputs high-level planning instructions such as LEFT, RIGHT. An end-to-end model (a neural network) then takes the sensor data and the instruction and outputs actions such as steering and throttle.

Fig. 1 demonstrates the key idea of our architecture in an environment where the ego vehicle encounters obstacles in a corridor. As shown in Fig. 1, the end-to-end model is only trained with an environment consisting of only a single front obstacle. The ego vehicle learns to steer left or right to avoid obstacles. The end-to-end model might not recognize additional obstacles in adjacent locations. In this scenario, LLMs identify new objects and generate instructions by selecting a wide clearance space to navigate around obstacles. During the evaluation stage, these instructions are used to guide the end-to-end model even in scenarios outside its training dataset. In this way, we enhance the generalization and robustness of the end-to-end model with LLMs. See Section 3.1 for more details about this experiment.

Our architecture employs LLMs without fine-tuning. Instead, we utilize the *Chain-of-Thought* (CoT) prompting method developed by Wei et al. [35]. In contrast to the previous work, our LLM focuses on generating high-level instructions (left, right, middle, etc) instead of individual actions (steering, throttle, etc). In doing so, we efficiently leverage the inherent abilities of LLMs for scenario understanding and reasoning and avoid the weakness of the LLM on specific calculations and inference latency. Conversely, generating actions would require fine-tuning the LLM on specific driving scenarios, resulting in data collection in complex environments. Another advantage of this design is that the LLM is allowed to run slower than the end-to-end model. The vehicle does not need to wait for each inference from the LLM to make a single action by caching and applying the previous LLM command to the end-to-end model before generating the next instruction. The end-to-end model uses a lightweight neural network that runs on edge devices or smartphones (for example, an iPhone in our robotic vehicle) and responds in milliseconds, meeting real-world inference constraints and compensating for the slow response of the LLM.

In summary, we propose an architecture for autonomous driving that combines end-to-end autonomous driving methods with LLMs. This architecture leverages the contextual understanding of LLMs to provide high-level instructions, enhancing the generalization and robustness of the end-to-end model. This combined model mimics human drivers who occasionally make deliberate decisions to alter driving patterns while following decisions instinctively with so-called muscle memory most of the time. The main contributions are as follows.

- We are the first to integrate LLMs with end-to-end autonomous driving and apply it in a real-world, closed-loop environment.

- Our framework utilizes a hybrid architecture that combines the rapid response capabilities of a small end-to-end model to achieve millisecond latency with the comprehensive

capacity of LLM for world understanding and inference, enabling great adaptation at the in-vehicle edge and embedding devices to new and previously inexperienced environments.

- We only require the LLM to generate high-level instructions rather than low-level actions. Our experiment confirms that this approach relaxes the need to fine-tune the LLM while achieving the generalization capability necessary for end-to-end driving.

- Our architecture only requires a small amount of training data for the end-to-end model that is easy to collect solely from vision-based sensors, such as vehicle-mounted cameras and smartphones.

**Related works:** *End-to-end autonomous driving*: End-to-end autonomous driving [13, 36] has flourished since NVIDIA first introduced it [37]. Unlike traditional autonomous driving, end-to-end autonomous driving outputs actions directly from sensor data. There are several methods for end-to-end autonomous driving, such as the world model [38, 39], multi-sensor fusing [40, 20], trajectory based control [41, 42], and multi-task/imitation learning [43, 44, 16, 45, 46]. However, the end-to-end autonomous driving model usually suffers from weak generalization capability. To address the issue, recent works [22] have explored the potential of using LLMs in autonomous driving to improve the generalization capability of the end-to-end model.

*LLM for autonomous driving*: Most previous works applied LLMs for autonomous driving in an open-loop manner [27, 28], where the output actions or predicted trajectories are compared with experts without applying them to an environment. Recently, Shao et al. [31] explored LLM for autonomous driving in the closed loop. They applied a fine-tuned multimodal LLM that takes the raw image and outputs actions. The experiment is conducted in a simulated environment. Paul et al. [21] and Azarafza et al. [32] also developed methods for LLM in a closed-loop manner. However, these methods require the LLM to generate outputs for every action taken by the ego vehicle, demanding a quick response from the LLM. The authors tested their methods in simulated environments where the slow response problem can be overlooked. Instead, our proposed method does not require a quick response from the LLMs and is vetted in a real driving environment.

## 2 Methodology

In this study, we develop an autonomous driving system utilizing a car to navigate a hallway, relying solely on images taken by a monoscopic camera. Our system captures frontal view images with the onboard camera and preprocesses images before passing them to the driving model to output the steering and throttle actions to control the car. While traditional end-to-end models often face generalization challenges inherent in data-driven approaches, LLMs have shown promise in learning rich representations and contextual understanding from extensive datasets. Inspired by the recent success of LLMs, we incorporated an LLM as an assistant in our end-to-end framework to mitigate the generalization problem.

### 2.1 Proposed architecture

The proposed architecture consists of two components: an end-to-end model and a pretrained LLM that requires no fine-tuning. The end-to-end model processes front-view images and outputs the corresponding actions, while the LLM provides high-level instructions based on the given images. The end-to-end model is trained to respond to the environment efficiently following the high-level instruction. In Fig. 2, we use `ChatGPT-4o` [47] as an example, demonstrating how our model processes images from continuous camera streams and periodically receives high-level instructions from the LLM at intervals determined by the LLM's inference speed. Next, we discuss how to train an end-to-end model and combine it with LLMs.

**End-to-end model** The end-to-end model must make predictions in real time from the inputs of the image data and the instructions from the LLM. To make the end-to-end model suitable for taking both images alone and images with high-level instructions as input, we use the network architecture

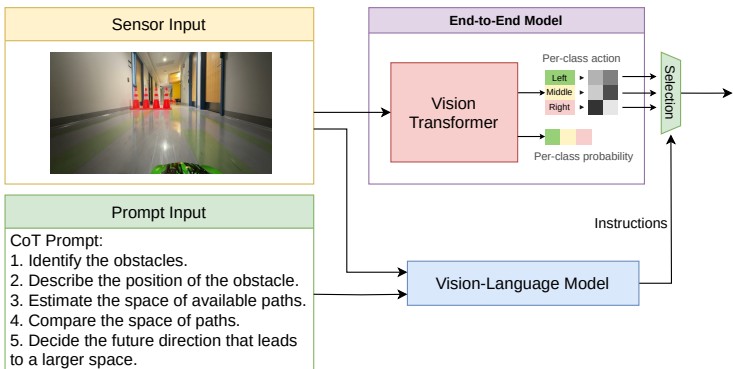

Figure 2: The proposed architecture inputs sensor data to both the LLM and the end-to-end model. The end-to-end model outputs actions from sensor images and receives slower, high-level instructions from the LLM due to its slower inference speed. This setup bridges the gap between fast-moving vehicles and the contextual insights and slow decisions from the LLM.

| High-level Query | Sub-queries |
|---|---|
| Decide the future direction. | 1. Identify any obstacle in the image.
2. Describe the position of the obstacles in the hallway.
3. Describe the position of the empty space between the obstacles and the wall along the hallway.
4. Describe which empty space is larger.
5. Output the direction of larger empty space as LEFT, MIDDLE, or RIGHT. |

Table 1: CoT query example

and training method similar to Hawke et al. [16] and Shafiullah et al. [48]. In Shafiullah et al. [48], the action space is clustered into $k$ different categories. Their model takes images as input and uses `MinGPT` to predict the categorical probability and the per-category action values.

In our end-to-end model, we employ a pre-trained Vision Transformer (ViT) [49] instead of using `MinGPT` as the image backbone. We manually configured the action space categories instead of using the learned k-means clustering because the action space in our architecture consists only of steering and throttle. Each category is assigned an LLM instruction. Details on the encoding of the action space can be found in Appendix B. We train the end-to-end model on the dataset containing the input images and actions taken by a human expert as labels, as proposed by Hawke et al. [16]. The end-to-end model requires only a minimal dataset including simple scenarios that illustrate the instructions provided by the LLM. Specifically, the end-to-end model is trained in an environment that contains only a single cone to avoid, and we use the planning capability of LLM to extend to more complex scenarios. More details about the training environment and the data collection process are discussed in Section 3.1.

**LLM for zero-shot inference:** As discussed earlier, our end-to-end model is lightweight and trained on limited or simple scenarios, resulting in a lack of generalization capability for more complex scenarios. To improve the generalization capability of the end-to-end model, an LLM is adopted to enhance the model's understanding of intrinsic scenarios by providing high-level instructions. Instead of fine-tuning the LLM, we utilize the prompt engineering technique CoT [35]. CoT breaks down complex tasks into sequential intermediate reasoning steps. Table 1 is an example of subqueries decomposed from a high-level instruction.

With the help of the LLM, the planning capacity to navigate multiple obstacles is integrated into the end-to-end model. The performance of LLM with and without CoT in our experiment is discussed in Appendix D.

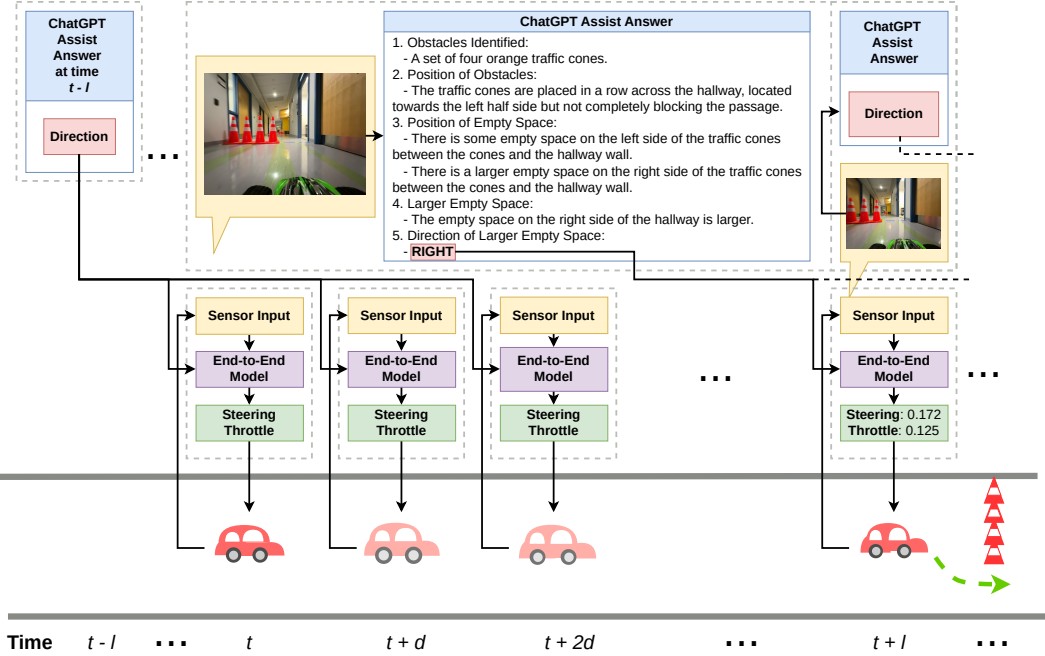

Figure 3: A closed-loop pipeline of the proposed architecture using `ChatGPT-4o`: The LLM takes the front-view image of the ego car with CoT prompts and generates the instruction. The end-to-end model then takes the previous LLM-assisted instruction, along with the real-time sensor input, and outputs steering and throttle for real-time control. The inference time of the end-to-end model and the LLM are denoted as $d$ and $l$ accordingly. The steering ranges from $-1$ (leftmost) to 1 (rightmost) and the throttle ranges between 0 to 1.

## 2.2 Closed-loop inference

The pipeline of our architecture in the closed-loop is demonstrated in Fig. 3 using `ChatGPT-4o` as an example. Since inference on the LLM takes longer than the end-to-end model, our end-to-end model uses the cached instruction from the previous LLM inference while waiting for the next instruction. This inference pipeline combines the world-knowledge of the LLM into the end-to-end model while still keeping the whole pipeline from suffering the slow inference speed of the LLM by making a lightweight end-to-end model that runs fast even on a smartphone.

## 3 Experiment

Our experiment aims to demonstrate that an LLM, even without fine-tuning, improves the generalization capability of an end-to-end model. Inspired by OpenBot [50, 51] and their early implementation [52], we conducted experiments in a real-world setting on a self-driving robot that integrates OpenBot on a commercial off-the-shelf RC vehicle with a smartphone as the embedded system onboard. We chose the iPhone in our hardware implementation because of its excellent support for `PyTorch`. The RC car can reach a maximum speed of 24 miles per hour. To avoid any hazard to people or property, we restricted its speed to less than ten miles per hour. The smartphone onboard sends action signals to the car that subsequently adjusts its steering and throttle accordingly. The only sensor in the car is the back camera of the attached phone. The end-to-end model also runs on the smartphone, enabling it to send low-latency action signals directly to the car.

### 3.1 Environment setup

To show the generalization property, we use two different environment setups to train the end-to-end model and test the proposed architecture discussed in Section 2.1.

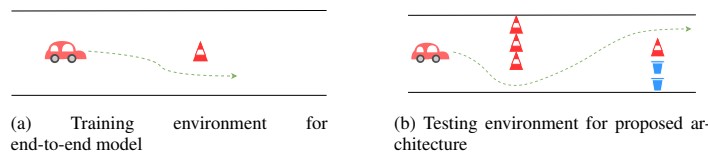

(a) Training environment for end-to-end model

(b) Testing environment for proposed architecture

Figure 4: Training and testing environments.

**Training environment for end-to-end models:** The training environment is designed to be as simple as possible. As shown in Fig. 4a[1], a cone is placed directly in front of the ego car. The ego car can choose to turn left or right to avoid the cone. We manually control the ego car to drive left or right, storing images at 60 frames per second as input and the corresponding actions as labels to train the end-to-end model. The training details can be found in Appendix B.

**Testing environment for proposed architecture:** As shown in Fig. 4b, several cones and trash bins are placed in a zigzag pattern in front of the ego car. To avoid a collision, the ego car must turn right first to avoid the cones blocking the hallway's left. After passing these cones, several more are placed right in front of the ego car, requiring the car to turn left to avoid a collision. This provides a challenging testing environment because the model has to reason and plan on what action to take to avoid the front obstacle without hitting the surrounding obstacles. Such a scenario never appears in the training dataset. Besides the large and small cones, the environment also includes various distracting items, such as trash bins and doors. The width of the road also changes, becoming wider after passing the initial set of cones. This provides a complicated testing environment to challenge the generalization capability of our proposed architecture.

## 3.2 LLMs and end-to-end models

We combine several LLMs with the end-to-end model. The LLMs we considered are `LLaVA-LLaMA2-13B` [53], `LLaVA-LLaMA3-8B` [54], `MiniGPT-v2` [55], and `ChatGPT-4o` [47]. The `ChatGPT-4o` runs on the OpenAI's server, while other LLMs run locally with two `NVIDIA RTX A6000` GPUs. The inference time of each model is summarized in Table 2. Note that the inference time required by `ChatGPT-4o` is even shorter than the inference time of some of the other models due to the superior computational resources provided by OpenAI compared to ours. The end-to-end model we utilized is `ViT-B/16` [49]. More details is discussed in Appendix B.

| Model Name | Inference time (s) |
|---|---|
| LLaVA-LLaMA2-13B | $7.76 \pm 0.56$ |
| LLaVA-LLaMA3-8B | $5.86 \pm 1.17$ |
| ChatGPT-4o | $7.09 \pm 2.80$ |
| MiniGPT-v2 | $7.30 \pm 1.24$ |

Table 2: Inference time of each LLM

## 3.3 Results

We evaluate our architecture based on different LLMs presented in Section 2.1 and Section 3.2 in the training and testing environments mentioned in Section 3.1. We also perform extended experiments on other types of obstacles and the experiment details and results can be found in Appendix C. We repeat 30 experiments in a real-world scenario, and the experiment is counted as "success" if the vehicle passes around all the cones without hitting any of the cones or the wall. The success rate is calculated by the number of successful experiments divided by the total number of experiments. The results of all experiments are summarized in Table 3. The experiments confirmed that the end-to-end

---

[1]The figure is only for demonstration; the actual environments are real-world settings, not simulations.

model can handle the scenario in its training dataset. However, for a more complicated environment shown in Fig. 4b, the end-to-end model only has a success rate of 40% because of a lack of training data on such scenarios. On the other hand, the LLM understands the complex environment and offers high-level guidance to steer the end-to-end model in the correct direction. Nevertheless, the LLM does not always generate correct instructions and experiences hallucinations when the environment is disrupted by rapidly changing lighting conditions. We will address these limitations of our driving model in Section 3.4.

| Model | Success Rate (%) | |
|---|---|---|
| | Train | Test |
| LLaVA-LLaMA2-13B + ViT | 100 | **83** |
| ChatGPT-4o + ViT | 100 | 75 |
| MiniGPT-v2 + ViT | 100 | 75 |
| LLaVA-LLaMA3-7B + ViT | 100 | 63 |
| ViT only | 100 | 40 |

$^\star$all values are rounded into decimal place.

Table 3: Comparison of different LLM models.

## 3.4 Limitations

Even LLMs without fine-tuning demonstrate the ability to enhance the generalization capability of the end-to-end model, but there are still limitations. First, in scenarios with strong backlighting and reflection in the front image, the LLM struggles to identify the position of obstacles, resulting in incorrect instructions. This issue arises because the dataset used to train the LLM contains few images with poor lighting conditions. Appendix D.1 provides examples of these failure cases. Second, to maximize the ability of the LLM in obstacle identification, the CoT prompt must be designed specifically for the obstacle avoidance task. A sophisticated prompt design and an LLM with a long contextual length are required to tackle complex driving tasks.

## 4 Conclusion

We develop an architecture that combines LLMs with an end-to-end model to improve the generalization capability of the end-to-end model. The key distinction between our architecture and previously proposed architectures is that the LLM in our architecture only provides high-level instructions rather than direct actions or trajectories. The end-to-end model uses both raw sensor data and high-level instructions from the LLM. These high-level instructions are received at a slower rate compared to the raw sensor data, allowing the end-to-end model to output actions even while the LLM still makes inferences. By integrating LLMs with the end-to-end model, our architecture relaxes data collection requirements and eliminates the need for rapid responses from LLMs. This design makes it feasible for real-world applications. Experiments are conducted in a real-world environment with a self-driving vehicle. The vehicle is equipped solely with the front-view camera as its sensor. We train the driving model in a simple yet common environment and deploy the model on the smartphone. The results show that the end-to-end model alone cannot navigate safely in a complex environment that does not appear in the training data. However, when combined with LLMs, the end-to-end model navigates the complex environment successfully. Notably, the LLMs we utilize are not fine-tuned; instead, we employ chain-of-thought prompt engineering techniques to enhance performance. The results demonstrate that LLMs improve the generalization capability of the autonomous driving system, enabling it to adapt to more complex environments with simple training data. However, LLMs still exhibit limitations, such as susceptibility to hallucinations when the environment changes, such as backlighting and ground reflection. This requires further investigation and a deeper understanding of LLM behavior.

**Acknowledgments**

This material is based upon work supported by the U.S. Department of Energy, Office of Science, Office of Advanced Scientific Computing Research, Office of Nuclear Physics, under Award Number DE-SC0019518.

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

# A  Limitations of end-to-end network on extending to new scenarios

Training an end-to-end model requires a massive amount of data to enable the model to adapt to new environment settings, such as varying objects on the road, weather and lighting conditions, road textures, and colors. Without a sufficient and diversified dataset, it faces challenges when extending from simple to complex scenarios (not shown in the training dataset) that involve planning and reasoning. In this section, we illustrate this limitation using a simple yet representative experiment.

In the experiment, we consider a three-lane driving scenario, where the lanes are designated as left, middle, and right. We create different scenarios by placing obstacles in one or more lanes, denoted as to $S_{a_l a_m a_r}$, where $a_l$, $a_m$, and $a_r$ represent the presence of an obstacle in the left, middle, and right lanes, respectively (with 1 indicating an obstacle and 0 indicating no obstacle). The training dataset is collected in the scenario $S_{010}$, which represents an obstacle in the middle lane. In this scenario, the car is trained to navigate around the obstacle by using either the left or right lane. We then evaluate the model on scenarios $S_{010}$, $S_{110}$, and $S_{011}$.

| Scenario | Success rate (%) |
|----------|------------------|
| $S_{010}$ | 100 |
| $S_{110}$ | 44 |
| $S_{011}$ | 78 |

Table 4: Experiment results on out of sample scenarios.

As shown in the experiment result in Table 4, the end-to-end model is not able to reliably extend to even a slightly modified scenario. Although it is possible to include more scenarios like $S_{110}$ and $S_{011}$ in the training dataset, there will always be edge cases that remain unaccounted for. This is why we introduce the VLM to provide a "world model" that enhances the planning and reasoning capabilities of the end-to-end model, enabling it to generalize from simple to complex scenarios by decomposing complex behaviors into a set of atomic actions.

# B  End-to-end network details

This section includes the implementation details of the end-to-end model discussed in Section 2.1. The end-to-end model is a neural network. The neural network takes the image and instructions from the LLM and outputs the steering and throttle actions. The input image is the front view of the car and is taken by the ultra-wide camera of the `iPhone 15 Pro` mounted on the car. The field of view of the camera is 101 degrees, and the raw image size is $640 \times 480$. The image is further cropped with a rectangular area $[\texttt{top}, \texttt{bottom}, \texttt{left}, \texttt{right}] = [140, 330, 130, 510]$ and is resized to $320 \times 160$ before being sent to the end-to-end model. The input instruction is a set of commands $\{\texttt{LEFT}, \texttt{MIDDLE}, \texttt{RIGHT}\}$, indicating which direction the car should go to avoid the front obstacles if the input image indicates there are any. Also, to simplify the response to the LLM, we ignore any cases and recognize `STRAIGHT` as `MIDDLE`.

## B.1  Neural network architecture

The image backbone for the end-to-end model is the pretrained `ViT-B/16` model provided by Dosovitskiy et al. [49]. The model is modified according to Shafiullah et al. [48] to include two prediction heads: one outputs the per-class action values, and the other outputs the classification probability. The per-class action-value output $V$ is a matrix of size $3 \times 2$, where each row indicates one of the three commands in the instruction set, the first column is the steering value, and the second column is the throttle value. The classification probability output $p$ is the probability density for each class.

## B.2 Dataset collection

The dataset consists of 60 different routes. Each route consists of a sequence of action pairs ($\texttt{image}, \texttt{steering}, \texttt{throttle}$) with an average length of 100. To train the end-to-end model to take the three different instructions $\texttt{LEFT}$, $\texttt{RIGHT}$, and $\texttt{MIDDLE}$, we collected three different types of routes in the training environments illustrated in Fig. 4a: the first and the second are to pass around the obstacle from the right and left, respectively, and the third is collected when there are no obstacles in the front and the car goes straight. Each route is labeled by its route type. We will discuss in the next subsection how these data are used to train the end-to-end network for different instruction outputs.

## B.3 Training the end-to-end model

As discussed in Section B.1 and Section B.2, the neural network outputs three different actions, and the dataset also contains different types of route behaviors. In other words, each sample in the dataset is considered the tuple $D = (X_{img}, y_s, y_t, y_c)$, where $X_{img}$ is the input image, $y_s$ is the steering value range $[-1, 1]$, $y_t$ is the throttle value range $[0, 1]$, and $y_c$ is the route label selected from the set $\{1, 2, 3\}$, representing the three instructions. The neural network can be written as $(V, p) = f(w; X_{img})$, where $V$ is a matrix of $3 \times 2$ representing the per-class action-value output, $p$ is a vector of 3 representing the probability distribution of each class, and $w$ is the weight of the neural network. Then the end-to-end model is trained with the supervised learning method with the loss function defined as

$$L(w; D) = (V_{y_c,0} - y_s)^2 + (V_{y_c,1} - y_t)^2 - k \log p_{y_c},$$

where $w$ is the hyper-parameter to weight the loss of the action value and the cross-entropy loss of the classification.

## B.4 Evaluation

We evaluate the end-to-end model in real-world, real-time, and closed-loop conditions. We conduct experiments with and without LLM assistance instructions. When high-level instructions and images are used as input, the prediction probability head is discarded, and the LLM provides the steering and throttle values. When only images are used as input, the actual instruction is sampled from the predicted probability head.

# C   Extended experiment: moving obstacles

We extend our experiment to more complex scenarios where moving obstacles (e.g. cars, pedestrians) are present. The settings for training and testing are similar to Fig. 4a and Fig. 4b. Instead of placing cones in the left or right lane, the obstacle is replaced with moving pedestrians or moving cars. We performed 10 repeated experiments on these testing scenarios, using both an online VLM (ChatGPT-4o) and a local VLM (LLaVa-LLaMA2). The success rates for each scenario are shown in Table 5.

| VLM | Scneario 1 (Car) | Scneario 2 (Ped) | Scenario 3 (Car+Ped) |
|---|---|---|---|
| ChatGPT-4o | 50% | 60% | 33% |
| LLaVA-LLaMA2 | 100% | 80% | 55% |

Table 5: Comparison of different LLM models with moving obstacles.

# D   Prompt engineering and results

This section compares the performance across different prompting techniques. Table 6 shows the query context using naive prompting and CoT prompting. We use two different types of query

| Prompt Type | Context |
|---|---|
| Naive | The image shows a toy car drives through a hallway that might have obstacles. Please output the future direction of the car as LEFT, MIDDLE, or RIGHT. |
| CoT | A toy car drives through a hallway that might have obstacles. Please answer the following 5 questions step by step: 1. Identify any obstacle in the image. 2. Describe the position of the obstacles in the hallway. 3. Describe the position of empty space between the obstacles and the hallway wall. 4. Describe which empty space is larger. 5. Output the direction of larger empty space as LEFT, MIDDLE, or RIGHT. |

Table 6: Prompt content for naive, CoT

images where the car should go left or right to avoid the obstacles. Fig. 5a and Fig. 5b show examples of query images. Tables 7 - 10 are sample responses for Fig. 5a. Tables 11 - 14 are sample responses for Fig. 5b.

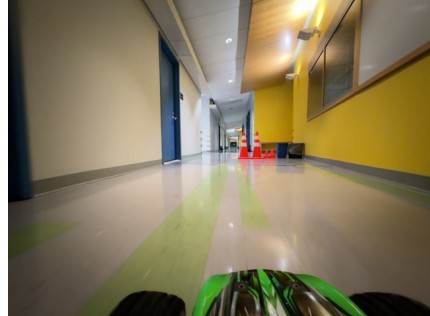 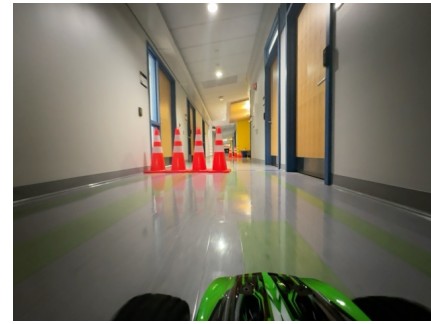

(a) Ground truth instruction: LEFT.    (b) Ground truth instruction: RIGHT.

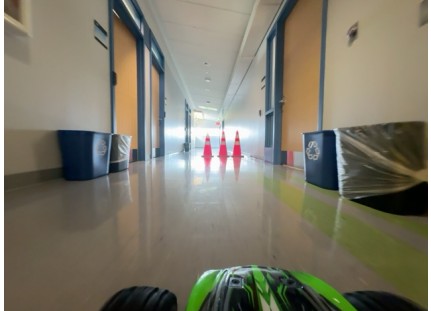

(c) Example of images with backlighting and reflections.

Figure 5: Example images.

## D.1 Limitations

This section discusses the case where the LLM might fail. Fig. 5c shows an example image where the image contains intensive backlight and reflects light from the ground. In that case, the LLM may fail to generate the correct answer. Table 15 shows a sample output for Fig. 5c using LLaVA-LLaMA2.

| Prompt Type | Response |
| --- | --- |
| Naive | Based on the image, the toy car appears to be moving towards the **right** side of the hallway. |
| CoT | 1. There are orange traffic cones in the hallway, which are obstacles.
2. The traffic cones are positioned in the middle of the hallway, creating a pathway on either side.
3. The empty space between the obstacles and the hallway wall is on both sides of the cones, with the space on the left side being larger than the space on the right side.
4. The car should drive on the right side of the obstacle, as the space on the right is smaller and would be more challenging to navigate.
5. The car should drive on the **left** side of the obstacle, as the space on the left is larger and would provide a smoother path. |

Table 7: Response of prompts for Fig. 5a by LLaVA-LLaMA2

| Prompt Type | Response |
| --- | --- |
| Naive | Based on the image, the toy car is currently in the middle of the hallway. It is **not possible** to determine the exact direction the car will take without more information or a clearer view of the obstacles in the hallway. |
| CoT | 1. Yes, there is a cone in the hallway.
2. The cone is located in the middle of the hallway.
3. The empty space between the cone and the hallway wall is larger on the right side.
4. The car should drive on the **left** side of the cone. |

Table 8: Response of prompts for Fig. 5a by LLaVA-LLaMA3

| Prompt Type | Response |
| --- | --- |
| Naive | **RIGHT**. |
| CoT | 1. There are two obstacles in the hallway, one on the left and one on the right.
2. The obstacles are positioned parallel to each other and are spaced about 3 feet apart.
3. The empty space between the obstacles and the hallway wall is approximately 6 feet long.
4. The empty space on the left side of the obstacles is larger than the empty space on the right side.
5. The car should drive on the **left** side of the obstacles to avoid hitting them. |

Table 9: Response of prompts for Fig. 5a by MiniGPT-v2

| Prompt Type | Response |
|---|---|
| Naive | The future direction of the toy car should be to the **LEFT**. The hallway ahead is obstructed by traffic cones which requires the car to navigate around them. |
| CoT | Step 1: Identify any obstacle in the image
- The image shows multiple red traffic cones as obstacles in the hallway.
Step 2: Describe the position of the obstacles in the hallway
- The traffic cones are positioned in the middle and slightly to the right of the hallway.
Step 3: Describe the position of empty space between the obstacles and the hallway wall
- There is empty space both on the left side and the right side of the obstacles. The left side is between the obstacles and the left wall, and the right side is between the obstacles and the right wall.
Step 4: Describe which empty space is larger
- The empty space on the left side between the obstacles and the left wall is larger compared to the space on the right side.
Step 5: Output which side of the obstacle the car should drive as LEFT, MIDDLE, or RIGHT
- The car should drive on the **LEFT** side of the obstacles. |

Table 10: Response of prompts for Fig. 5a by ChatGPT-4o

| Prompt Type | Response |
|---|---|
| Naive | Based on the image, the toy car appears to be moving towards the **right** side of the hallway. |
| CoT | 1. There are orange traffic cones in the hallway, which are obstacles.
2. The traffic cones are positioned in a line along the hallway, with the first cone closer to the camera and the last cone further away.
3. The empty space between the cones and the hallway wall is larger on the right side of the image.
4. The empty space on the left side of the image is smaller.
5. The car should drive on the **right** side of the obstacle, which is the side with the larger empty space. |

Table 11: Response of prompts for Fig. 5b by LLaVA-LLaMA2

| Prompt Type | Response |
|---|---|
| Naive | Based on the image, the toy car is currently in the middle of the hallway, and there are no visible obstacles in the immediate path. Therefore, the future direction of the car is likely to be **straight** ahead. |
| CoT | 1. Yes, there are orange and white cones in the hallway.
2. The cones are lined up in a row, with one cone placed in the middle of the hallway and two cones placed on either side of the hallway.
3. The empty space between the cones and the hallway wall is larger on the right side of the hallway.
4. The car should drive through the larger empty space on the right side of the hallway.
5. The car should drive through the larger empty space on the **right** side of the hallway. |

Table 12: Response of prompts for Fig. 5b by LLaVA-LLaMA3

| Prompt Type | Response |
|---|---|
| Naive | **RIGHT**. |
| CoT | 1. There are two orange cones in the hallway.
2. The obstacles are placed in a zigzag pattern.
3. The empty space between the obstacles and the wall is approximately 10 feet long.
4. The left side of the obstacle is closer to the wall.
5. The car should drive on the **right** side of the obstacle. |

Table 13: Response of prompts for Fig. 5b by MiniGPT-v2

| Prompt Type | Instruction |
|---|---|
| Naive | To avoid the obstacles (traffic cones) in the hallway, the toy car should move to the **LEFT**, as it is the most clear path available. |
| CoT | 1. **Identify any obstacle in the image:**
- The obstacles are orange traffic cones placed in the hallway.
2. **Describe the position of the obstacles in the hallway:**
- The traffic cones are positioned almost in the middle of the hallway but towards the left side slightly.
3. **Describe the position of empty space between the obstacles and the hallway wall:**
- There is a significant empty space on the right side of the hallway next to the traffic cones.
- There is a smaller empty space between the traffic cones and the left wall of the hallway.
4. **Describe which empty space is larger:**
- The empty space on the right side of the traffic cones is larger than the empty space on the left side.
5. **Output which side of the obstacle the car should drive as LEFT, MIDDLE, or RIGHT:**
- The car should drive on the **RIGHT** side of the obstacle. |

Table 14: Response of prompts for Fig. 5b by ChatGPT-4o

| Prompt Type | Instruction |
| --- | --- |
| Naive | The image shows the toy car approaching three traffic cones placed in the middle of the hallway, partially blocking the middle path. Given the position of the obstacles, the future direction of the car should be **RIGHT**. |
| CoT | To analyze the provided image of a toy car driving through a hallway with obstacles, let's address the questions step by step:
1. **Identify any obstacle in the image:**
- The obstacle in the hallway is a set of three orange traffic cones placed in a row.
2. **Describe the position of the obstacles in the hallway:**
- The traffic cones are positioned in the center of the hallway, forming a barrier that extends across the middle part of the hallway's width.
3. **Describe the position of the empty space between the obstacles and the hallway wall:**
- There is empty space to the left and right of the traffic cones. The left side of the hallway (from the perspective of the toy car) has some space between the cones and the wall with trash/recycling bins. The right side also has space, but there are bins near the wall as well.
4. **Describe which empty space is larger:**
- The right side of the obstacle appears to have a slightly larger empty space compared to the left side. The right side's space seems more unobstructed, especially considering the positioning of the bins which seem to be slightly farther apart or away from the wall, giving a bit more clearance.
5. **Output which side of the obstacle the car should drive as LEFT, MIDDLE, or RIGHT:**
- The car should drive to the RIGHT side of the obstacles where there is more room.
So, the output is: **RIGHT**. |

Table 15: Response of prompts for Fig. 5c by ChatGPT-4o

