# OpenReview forum: "Generalizing End-To-End Autonomous Driving In Real-World Environments Using Zero-Shot LLMs"
_robot-learning.org/CoRL/2024/Conference — CoRL 2024_

### Official Review · Reviewer_Fw6g · 2024-07-20
**A solid step towards applying LLM4AD**

**Originality:** 3
**Technical Quality:** 4
**Clarity Of Presentation:** 3
**Potential Impact:** 3
**Recommendation:** 4
**Confidence:** 5

**Review:**

Applying LLM for autonomous driving has been widely discussed in the community. However, there are several common concerns:
1. Latency: LLM is heavy while autonomous driving requires high FPS decision making to ensure safety.
2. Numerical: LLM has inherent issues to handle numerical inputs and outputs due to the tokenization process, while in autonomous driving, the vectorized world and control trajectories/signals are numerical.
3. Necessity: If we simply use LLM for regression, it serves as an huge MLP, which could not utilize their reasoning ability well.

The authors solve them by only using LLM with raw sensor inputs for high-level instruction while using an end-to-end ad model for actual control.  In this way, LLM dose not need to process numerical inputs and outputs and the high-level instruction could be more naturallly integrated into language to utilize LLM's reasoning ability. At the same time, the adoptation of another e2e-ad model avoid the response requirements for LLM at every step.

Finally, they deploy their methods in the real world, which fits CoRL well.

**Quality Of The Limitations Section:**

3

**Questions For Rebuttal:**

Missed reference in the end-to-end autonomous driving field:
1. Trajectory-guided Control Prediction for End-to-end Autonomous Driving: A Simple yet Strong Baseline. NeurIPS 22
2. Think Twice before Driving: Towards Scalable Decoders for End-to-End Autonomous Driving. CVPR 23
3. DriveAdapter: Breaking the Coupling Barrier of Perception and Planning in End-to-End Autonomous Driving. ICCV 23 Oral
4. Think2drive: Efficient reinforcement learning by thinking in latent world model for quasi-realistic autonomous driving (in carla-v2). ECCV 24

**Robotics Focus:**

4

**Summary Of Paper:**

This work explores several practical issues of deploying LLM for autonomous driving. They solve it by using LLM for high-level instruction and using an e2e model for low-level control. They deploy their algorithm in the real world.

**Summary Of Recommendation:**

To apply LLM for actual autonomous driving, the data issue and latency issue are non-negligible. This work starts an initial discussion and deploy their algorithm in the real world, which fits CoRL well. Thus, I give strong accept.

---

### Official Review · Reviewer_1ZPV · 2024-07-20
**Initial Review**

**Originality:** 3
**Technical Quality:** 2
**Clarity Of Presentation:** 3
**Potential Impact:** 2
**Recommendation:** 2
**Confidence:** 4

**Review:**

Strengths:
- While I believe the experimental setting is too simple to be referred to as "autonomous driving", it is nonetheless impressive to see the system tested in a real world, closed-loop setting with LLM inference run online.
- Using an LLM for high level reasoning along with a neural network for lower level controls is intuitively well-motivated.

Weaknesses:
- The **main weakness** of the paper is that the proposed method is built for a toy setting of navigating static obstacles in a particular indoor environment. It's unclear if the proposed approach has much practical value for a real setting with dynamic obstacles. Even without obstacles, real world driving would likely require more complex commands than LEFT, RIGHT, MIDDLE to be provided by the high level reasoning system.
- The related works makes at least one incorrect statement: claiming NVIDIA introduced end-to-end autonomous driving (lines 121-122). The NVIDIA paper itself references ALVINN as the first end-to-end autonomous driving system introduced by Pomerleau in 1989. The section also misses important relevant works in end to end autonomous driving such as [1].
- The paper contains a reasonable amount of typos or grammatically incorrect sentences. Also, the word MIDDLE/STRAIGHT seem to be used interchangeably when referring to the command, which adds complexity for the reader. I would recommend additional proofreading for the final submission for improved clarity.

[1] End-to-end Interpretable Neural Motion Planner, Zeng et al. CVPR 2021.

**Quality Of The Limitations Section:**

2

**Questions For Rebuttal:**

- How would you extend the proposed method to support real world autonomous driving?
- Why do you believe certrain LLMs outperform others? Is it well correlated with the performance of those LLMs on other tasks?

**Robotics Focus:**

4

**Summary Of Paper:**

This paper presents a system for autonomous driving that leverages a multi-modal LLMs for high level reasoning along with an command-conditioned end to end model to execute low level controls. More specifically, an off-the-shelf LLM is given the image along with a chain-of-thought prompt to identify obstacles and provide one of three directions: left, right, middle.  The given command is then given to the end-to-end model which predicts the controls. The paper demonstrates the effectiveness of this approach in a real-world closed loop, but toy setting of a indoor robot with a configuration of obstacles.

**Summary Of Recommendation:**

While I appreciate the real world experiments, I believe the approach and setting are too simplified for the paper to provide much value to the research community.

---

### Official Review · Reviewer_XRbz · 2024-07-21
**Review of Submission 257**

**Originality:** 2
**Technical Quality:** 2
**Clarity Of Presentation:** 3
**Potential Impact:** 2
**Recommendation:** 2
**Confidence:** 4

**Review:**

Strengths:
 - Effectively applies CoT reasoning to a robotics problem
 - Identifies an issue with just directly applying VLMs as controllers (inference latency) and proposes a hybrid architecture to rectify it.

Weaknesses:
 - The CoT prompt is highly specific to this exact scenario and likely could not (in its current form) reason through more realistic open-world challenges like pedestrian/vehicle interactions or semantic scene understanding
 - Only evaluated in relatively simple obstacle-avoidance scenarios.
 - Experimental evaluations and analysis are severely lacking; at the very least comparisons to stronger driving models and scaling results with additional data should be included.
 - Having only one train and one test scenario limits claims of generalizability.
 - The work has limited novelty; i.e. it is a fairly straightforward application of CoT-conditioning to a language-conditioned policy at the low level. Experimental results are not sufficiently rigorous or impressive to make up for this lack of novelty.

**Quality Of The Limitations Section:**

3

**Questions For Rebuttal:**

- It is claimed that "even if cones replace trash bins, the car may still crash into the cones because one possible control pattern it learned from the training data dictates turning left when encountering cones". How does the VLM+E2E hybrid architecture improve this? Are we assuming that the underlying E2E policy places more weight on the text instruction than on the image? What happens when the image and text instruction conflict?
 - Does the E2E model express any actual capabilities other than directly following what the LLM tells it to do? Perhaps a good additional baseline would be to directly control the car by the VLM's outputs (+some underlying "blind" controller) to demonstrate the importance of the E2E policy.
 - When is the VLM queried? In the right-left testing environment, does the E2E policy know that it should wait for the VLM's response before executing the second task, or is the robot just "tuned" to move slowly enough that this is not a problem? If the latter, does that mean the robot will be unable to react if the tow rows of obstacles are placed closer together?
 - Could the CoT query be generalized so as to be usable for more complex reasoning tasks? It would be nice to have a baseline with a more open-ended CoT task.

**Robotics Focus:**

4

**Summary Of Paper:**

Use chain-of-thought reasoning to drive a car in OOD settings

**Summary Of Recommendation:**

Based on the limited novelty and weak experimental results I would recommend rejection. The authors would need to provide experimental results in significantly more complex scenarios, and significantly increase rigor and depth of analysis, for me to change my recommendation.

---

### Author Rebuttal · Authors · 2024-08-12

Attached supporting videos and the updated manuscript.

---

### Decision · Program_Chairs · 2024-09-04

**Decision:**

Accept

**Comment:**

In brief, the strengths of the paper are, the use of LLM in autonomous driving, effective reasoning strategy, and end-to-end approach. Although there were concerns, during the rebuttal stage the authors have successfully addressed the weaknesses of the papers to clarify and significantly improve the quality of the publication.